# Regulatory B Cells Involvement in Autoimmune Phenomena Occurring in Pediatric Graves’ Disease Patients

**DOI:** 10.3390/ijms222010926

**Published:** 2021-10-10

**Authors:** Kamil Grubczak, Aleksandra Starosz, Karolina Stożek, Filip Bossowski, Marcin Moniuszko, Artur Bossowski

**Affiliations:** 1Department of Regenerative Medicine and Immune Regulation, Medical University of Bialystok, 15-269 Bialystok, Poland; aleksandra.starosz@umb.edu.pl (A.S.); marcin.moniuszko@umb.edu.pl (M.M.); 2Department of Pediatrics, Endocrinology and Diabetes with a Cardiology Unit, Medical University of Bialystok, 15-274 Bialystok, Poland; karolina.stozek33@gmail.com (K.S.); felicjian@gmail.com (F.B.)

**Keywords:** regulatory B cells, Graves’ disease, methimazole, autoimmunity, thyroid, AITD

## Abstract

Graves’s disease is the most common type of autoimmune hyperthyroidism. Numerous studies indicate different factors contributing to the onset of the disease. Despite years of research, the exact pathomechanism of Graves’ disease still remains unresolved, especially in the context of immune response. B cells can play a dual role in autoimmune reactions, on the one hand, as a source of autoantibody mainly targeted in the thyroid hormone receptor (TSHR) and, on the other, by suppressing the activity of proinflammatory cells (as regulatory B cells). To date, data on the contribution of Bregs in Graves’ pathomechanism, especially in children, are scarce. Here, we investigated the frequencies of Bregs before and during a methimazole therapy approach. We reported higher Foxp3+ and IL-10+ Breg levels with CD38- phenotype and reduced numbers of CD38 + Foxp3 + IL-10+ in pediatric Graves’ patients. In addition, selected Breg subsets were found to correlate with TSH and TRAb levels significantly. Noteworthy, certain subpopulations of Bregs were demonstrated as prognostic factors for methimazole therapy outcome. Our data demonstrate the crucial role of Bregs and their potential use as a biomarker in Graves’ disease management.

## 1. Introduction

The most up-to-date data indicate that around 2–3% of people worldwide are suffering from Graves’ disease, which is the leading cause of hyperthyroidism [1]. Statistics indicate its highest prevalence among women at a ratio of 5–10:1 versus men. Considering our own observations, these numbers seem relevant, even in the context of pediatric patients [2,3]. Numerous factors were described as predisposing to autoimmune thyroid disease (AITD), including Graves’ disease. Among these, specific autoantibodies against TSH receptor (TSHR) can be distinguished, together with dysfunction of numerous immunoregulators leading to lack of immune reaction control or excessive activation of cells. Those immunomodulators include inter alia: Foxp3, CTLA-4, HLA-DR3, CD25, CD40, and, together with anti-TSHR autoantibodies, contribute to about 70% of AITD [2].

Graves’ disease, a highly complex condition as a result of an autoimmune background, is associated with the participation of numerous immune cell populations, including those of greatest popularity recently: regulatory T cells, Th17 lymphocytes, NK cells, plasmocytes, immunosuppressive B cells, dendritic cells [4,5,6,7]. B cells play essential roles in autoimmune reactions occurring in the course of Graves’ disease, not only as a source of autoantibodies against thyroid-stimulating hormone receptor (TRAb), but also through regulation of immune responses as regulatory B cells [5]. Immunosuppressive B lymphocytes participate in inflammation by inducing regulatory T cells and supporting iNKT maintenance and suppression of proinflammatory cells, such as TNF-alpha-producing monocytes, Th17 and Th1 lymphocytes, and cytotoxic CD8+ T cells [8]. B cells were found to be a dominant population of lymphocytes in the hyperactive thyroid specimens. In addition, almost 78% of samples positive for CD20 marker demonstrated concomitantly a high expression of Foxp3 predominantly. Considering low scores detected for CD4 T cells, these Foxp3 events detected might be presumably associated with regulatory B cell function [9]. Despite the essential role of regulatory B cells, it is worth noting that, in some autoimmunological conditions, including systemic lupus erythematosus (SLE), these cells clearly demonstrated a deficit in immunosuppressive properties [10].

Production of IL-10 immunosuppressive cytokine is a well-known marker used to assess regulatory B cells [11]. Its role was essential in reference to Bregs-mediated reduction in effector immune cells function and promotion of immunosuppressive populations [12]. Among all factors closely related to regulatory function, several surface markers are also considered significant for Bregs, including CD5, CD24, CD27, and CD38 [13]. CD38 was demonstrated to be highly expressed within the population of CD1d+CD5+ B cells, and, thus, is suggested to be one of the regulatory B cell markers. However, through investigation of CD38-deficient mice, the same study revealed that the presence of CD38 is not necessary for proliferation of cells with immunosuppressive potential, demonstrating even higher numbers of Bregs in that model [14]. Furthermore, despite cocultures of human activated CD4+ T cells with CD24+CD38− B cells leading to lower inhibition rates compared to CD38-positive cells, these subsets still maintained immunosuppressive activity with IFN-gamma and TNF-alpha release reduction [10]. Besides IL-10 and CD38 used in that study for Bregs discrimination, transcription factor Foxp3 has also been suggested as a marker for the regulatory potential of B cells [15]. Despite differences in Bregs assessment and various identification options, the presence of immunosuppressive markers and cytokines remains unquestionably the most essential element in confirming immunoregulatory potential of studied Breg subsets [11,16].

Methimazole (MMI) application, as an antithyroid drug (ATD), is one of the dominant therapeutic options in the management of Graves-related thyroid gland hyperactivity. It is preferred, especially in newly diagnosed patients, for both adults and children [1]. Despite being a relatively noncomplex compound aimed at blocking of iodination of thyroglobulin tyrosine residues, MMI also exerts other significant effects. Among these, immunosuppression plays a dominant role [17]. Noteworthy, these immune-related aspects were first described years ago [18], but our knowledge on methimazole influence is still incomplete. Its beneficial effects were described not only in the context of proper immune response restoration, but were also found to preserve the function of cells with regulatory function. Both immune response and mechanisms of its control can be adversely affected, for example, in the course of radioactive iodine (RAI) use [19,20]. Noteworthy, considering significantly lower remission rates observed in children [21,22], studies aimed at immunological aspects of ATD use might provide potential background for improving therapeutic efficacy.

Considering the lack of comprehensive research assessing the involvement of the Bregs in Graves’ disease, in pediatric patients predominantly, the aim of our study was to investigate changes in the Bregs subpopulation in Graves’ pediatric patients. Here, we decided to focus on the pediatric patients, considering the essential role of thyroid function on proper children’s growth and development and how disturbances in that gland might affect adulthood of these subjects. In addition, it is worth noting that the pediatric form of Graves’ disease demonstrates essential differences, even in blood morphology and immune cell distribution. That is associated inter alia with the fact that immune responses are at the developmental stage and mechanisms of tolerance are established. Therefore, an approach focused predominantly on children might not only reveal crucial aspects related to Graves’ disease pathogenesis, but also might allow for determination of the long-term effects—progression, response to therapy, life quality of adult subjects. Additionally, we also evaluated changes in the Bregs subsets during standard methimazole therapy to correlate these results with thyroid function parameters and establish patients’ Bregs phenotype with the best therapy outcome. Thus, the presented data shed a new light on Bregs participation in Graves’ disease and provide novel insight into their potential use in a diagnostic and clinical setting.

## 2. Results

### 2.1. Alterations in Levels of Circulating Regulatory B Cells from Graves’ Pediatric Patients Compared to a Healthy Control Group

Due to the lack of comprehensive data on regulatory B cell distribution in pediatric Graves’ patients, we initially decided to investigate the number of selected regulatory B cell subtypes between Graves’ versus the healthy control group. We did not observe significant differences in the context of IL-10- or Foxp3-expressing CD19+CD38+ B cells between studied groups at the admission stage. Compared to healthy controls, pediatric Graves’ patients showed higher numbers of regulatory B cells characterized by the presence of IL-10 or Foxp3 and lack of CD38 surface marker (CD38-negative) (Figure 1a,b). On the contrary, a decline in the level of B cells with immunosuppressive phenotype was demonstrated within CD19+CD38+ population with co-expression of both IL-10 and Foxp3 (Figure 1c). In addition to described changes, pediatric Graves’ patients revealed a decrease in C19+CD38− cells. (Figure 1d).

The percentage changes in the regulatory B cells within CD19-positive cells are in accordance with their absolute values. Higher frequencies compared to healthy controls were reported in the context of B cells with no CD38 marker expression and intracellular presence of IL-10+ or Foxp3+. Moreover, reduced values for CD38+ cells with co-expression of Foxp3 and IL-10 were reported in Graves’ disease pediatric patients. Despite the fact that absolute change of CD38+ B cells was not followed by CD38- cells, here, variations in both populations were mutually opposite. Interestingly, although the number of CD19+CD38− B cells was reduced in Graves’ patients, we reported a shift towards increased frequencies of CD38− B cells with a simultaneous decline in CD38-positive population (Figure 2a–d).

Considering the differences demonstrated above between Graves’ patients and the control group in the context of Bregs, we wondered whether associations between studied cell subsets are also affected in these groups. Interestingly, we found numerous significant links between studied populations of cells within CD19+ B cells. Foxp3+ and IL-10+ Bregs demonstrated a strong positive mutual correlation, and each individually showed significant association with populations based additionally on CD38 marker expression. However, none of the Foxp3+ or IL-10+ CD19+ cells correlated with CD38-negative B cells co-expressing Foxp3 and IL-10. Moreover, these subsets showed moderate/strong negative association with CD38+Foxp3+IL-10+ Breg subset. Considering expression of CD38, negative correlations with CD38+ Bregs co-expressing Foxp3 and IL-10 seemed to be related to CD38- cells with IL-10+/Foxp3+ rather than CD38+ cells. Noteworthy, Graves’ patients’ mutual dependencies were only to a narrow extent comparable to those observed in the healthy control group. Excluding all closely overlapping correlations, we found that healthy subjects’ Breg-related cell populations demonstrated a more complex network of mutual connections. IL-10+, CD38+IL-10+, CD38-/CD38+Foxp3+, and CD38+Foxp3+IL-10+ subsets were demonstrated to be the most involved participants. In contrast, Graves’ patients’ population of CD38+Foxp3+IL-10+ Bregs seemed to be the main regulator of other regulatory B cell subsets, importantly, in a negative manner. Importantly, the number of significantly different connections was clearly limited when compared to the healthy control group (Figure 2e,f).

### 2.2. Application of Methimazole Treatment Affects Distribution of B Cells with Regulatory Phenotype in Graves’ Disease Pediatric Patients

Treatment involving the use of methimazole caused essential changes within subsets of Breg in the course of Graves’ disease management. In most cases, crucial changes had appeared within the first 3 months of therapy. Although the population of CD38-Foxp3+ Bregs remained increased compared to healthy controls, its level was further elevated within the first months and restored to initial values at long-term observation. Interestingly, clear expansion of CD38+Foxp3+ B cells was also observed at first; however, further methimazole treatment decreased their level down to control values (Figure 3a). The number of IL-10-producing CD19+CD38−/+ cells remained unchanged during methimazole treatment. Only slight, statistically nonsignificant alterations were found after 1–2 years of MMI therapy, reaching values comparable to those of the healthy control group (Figure 3b). Furthermore, we found that methimazole also exerts its effects on CD38-Foxp3+IL-10+ B cell subset, with increasing values through all therapy, thus, concomitantly enlarging the difference between Graves’ and control subjects. Slightly different alterations were observed in CD38+ Bregs with Foxp3 and IL-10 co-expression. These cells achieved a level comparable to healthy controls after 3 months of the therapy; nonetheless, in long-term observation, that Breg subset was reduced back to pretreatment values (Figure 3c). Regarding the population of B cells expressing CD38 marker, the early increased number of CD38+ cells was later reduced after 1–2 years of therapy. In contrast, CD19+CD38- cells remained unchanged within the first 3 months, with a tendency for lower levels compared to the healthy control group (Figure 3d).

### 2.3. Treatment with Methimazole Influences Correlations between Regulatory B Cells and Thyroid Parameters of Graves’ Patients

When analyzing links between thyroid-related parameters and regulatory B cells of untreated Graves’ pediatric patients, the most significant correlations were reported in reference to TSH, fT3, and TRAb levels. In the context of TSH, we found a moderate negative correlation with CD38+Foxp3+ Bregs (*p* = 0.033), both in percentages and absolute numbers. Frequency of CD38+, even separately, was highly negatively correlated with TSH values (*p* = 0.034 for CD38+, *p* = 0.025 for CD38- B cell population). Furthermore, a moderate positive association was found between fT3 and the frequency of CD38-Foxp3+ in B cells (*p* = 0.047). Interestingly, a moderate negative correlation of fT3 with total CD38-negative cell numbers was reported. Regarding TSH-R antibodies, their level correlated negatively predominantly with CD38+ B cells; however, a weak association of the same direction was also reported in reference to the frequency of CD38+Foxp3+ Bregs. In addition to TRAb, quite the opposite correlation with the frequency of CD38+ B cells was observed compared to those found in TSH. Notably, the application of methimazole had a significant effect on mutual associations between Bregs and thyroid function parameters. A complete change in direction was shown, inter alia, in TSH versus Bregs correlations, with a moderate/strong correlation of TSH with both Foxp3+ and IL-10+ CD38-positive Bregs (*p* = 0.028 for CD38+Foxp3+ B cells). Similarly, correlation with CD38 presence was reversed from negative to strongly positive in the context of TSH level. Interestingly, despite the fact that the role of fT3 versus Bregs associations was reduced in the course of treatment, dominance of fT4 appeared to play a significant role at that point. In accordance, moderate positive correlations of fT4 were observed in reference to the frequency of CD38+Foxp3+IL-10+ Bregs and CD38-negative Foxp3+ and IL-10+ B cells. Moreover, correlations reported before treatment regarding TRAb and Bregs have also diminished in response to methimazole application. Cumulatively, most importantly, we found a significant shift from negative correlations of Bregs with TSH in untreated Graves’ patients towards moderately/strongly positive associations between these populations of regulatory cells and TSH together with fT4 (Figure 4).

### 2.4. Significance of Breg Profile in Graves’ Patients’ Response to Methimazole Application

Based on the Bregs assessment prior and in the course of treatment, we draw a hypothesis that the levels of regulatory B cells might have a crucial influence on methimazole-related effects on thyroid function in Graves’ pediatric patients. Studied subjects were stratified into groups demonstrating a low or high level of selected Breg populations and, subsequently, monitored in the course of methimazole treatment in the context of thyroid function—based on changes in TSH, fT3, fT4, and TRAb levels. First, we found that patients with high admission levels of Bregs with Foxp3+ and CD38 expression achieved higher TSH values in long-term observation compared to subjects with low numbers of that population. Such a difference was not observed in the context of subsets expressing IL-10 only. Interestingly, the observed phenomenon seemed to be closely related to the CD38 presence. In accordance, we found that a group with high CD38+Foxp3+IL-10+ cell numbers demonstrated the most significant increase in TSH level after 1–2 years of MMI treatment. In contrast, patients characterized by high numbers of CD38-Foxp3+IL-10+ B cells were found to respond less effectively to methimazole application after long-term observation. Despite the fact that such variations were not found in fT3 or fT4 levels, in most cases, patients with higher initial numbers of Bregs with CD38+ and Foxp3 and/or IL-10 demonstrated more pronounced changes in these thyroid-related parameters in response to treatment. Furthermore, as far as TSH-R antibodies are concerned, the intensity of alterations was dominant in patients with higher numbers of CD38+ Bregs, with obviously the opposite direction of changes compared to TSH levels. In addition, differences between groups of low and high Bregs were not only visible in subsets with Foxp3+ at the last time point of the treatment, but were also demonstrated for CD38-negative IL-10-expressing B cells, even at the early stage of the methimazole therapy. In general, considering this information, initial levels of regulatory B cells—those expressing Foxp3 and CD38 predominantly—might play a crucial role in response to methimazole treatment and, thus, contribute helpful biomarkers for the prediction of treatment effects (Figure 5).

## 3. Discussion

Regulatory B cells are well-known regulators of immune response and prevent excessive effector cell reactivity. Therefore, critical alterations in their function with inadequate or excessive activity have been linked to numerous autoimmune or neoplastic diseases, respectively [13]. However, the effort to understand the Bregs’ role in autoimmunity is hindered by its multiple phenotypes and functions. Therefore, a more recent purview on the Breg suggests focusing on the outcome of their interaction with immune cells and other surrounding elements of the environment instead of sticking tightly to the specific phenotypes [8,11,23]. Although being available in medical practice for more than 70 years, the complete mechanism of immunomodulatory properties of ATDs, including methimazole, is still missing [17]. In-depth investigation of the methimazole influence on immunity is of great importance in the face of its common use as a first selection in newly diagnosed nonpregnant Graves’ patients [1].

Lower values for IL-10+Foxp3+ Bregs are partially in accordance with our previous results on regulatory B10 cells in AITD, where reduction in Bregs frequency was reported in reference to CD19+CD24+CD27+IL-10+ B cells [24]. These data are also supported by reduced numbers of IL-10-producing B cells (B10 cells) in Graves’ disease presented by another research group [25]. Here, however, we found that additional identification using CD38 marker allowed B cell subsets demonstrating expression of Foxp3 of IL-10 to be revealed, with levels that are significantly elevated in Graves’ patients. These variations might indicate different roles of studied subsets in the autoimmune reactions associated with hyperactive thyroid, with an ambiguous nature of regulatory B cells. Interestingly, we found that, unlike CD38+Foxp3+IL-10+ Bregs, B cells without expression of CD38 also demonstrated the presence of Foxp3 or IL-10. Importantly, these populations were increased in the course of Graves’ disease. Our data might support previously demonstrated higher numbers of regulatory B lymphocytes in CD38−/− mice [14]. However, at the same time, further studies will be required to finally evaluate anti- or proinflammatory potential of these CD38-negative populations considering the observed phenomenon in children with Graves’. As the recent results showed even higher production of IL-10 in CD38−/− mice CD1d+CD5+ B cells compared to wild-type [14], we cannot assume that our strategy of delineation of CD38-negative cells is incorrect for the Breg assessment. In addition, despite the fact that CD38-negative CD1d+CD24+CD27+ Bregs were demonstrated to be less effective as immunosuppressors of effector T cell function, they still were able to reduce production of proinflammatory cytokines. Noteworthy, in autoimmunity associated with SLE, regulatory B cells with the highest immunosuppressive function demonstrated significantly diminished efficiency in inflammation control. In addition, such a phenomenon was not simply specific for autoimmune disorders in general (here reported in lupus erythematosus or AITD), as it was not shown in other conditions like Sjögren’s syndrome or osteoarthritis [10,26]. Presumably, expansion of CD38- B cells with Foxp3+ and/or IL-10 expression counteracts reduced numbers of CD38+Foxp3+IL-10+ Bregs and is a sign of a compensatory mechanism aimed at the reduction in autoimmune reactions related to Graves’ disease. In addition, such higher values for IL-10-producing B cells have already been reported in autoimmune phenomena associated with active rheumatoid arthritis [27]. However, subsequent investigation focusing directly on evaluated Bregs subset function is necessary to justify that hypothesis and reveal whether changes within regulatory B cells in the course of hypothyroidism are also associated with functional deficits in these cells.

Expression of CD38 was closely associated with autoimmunity development, as demonstrated in a mice model of lupus, thus suggesting its immunoregulatory role [28]. Interestingly, we found that CD38+ B cells correlated significantly with Breg populations only in healthy controls, while such associations were lost in the course of the autoimmune process. Our data demonstrated lower levels of regulatory B cells with CD38 expression and shifted towards a CD38-negative population. Thus, it should be established in the future whether observed changes are only compensatory mechanisms or are directly associated with Graves’ disease onset. Another option worth investigating is verifying whether reported CD38-negative B cells are separate subpopulations of cells or typical regulatory B cells with impaired function due to CD38 internalization or shedding and release as soluble form from the cell surface [29,30].

Differences in connections between CD38+ and regulatory B cells were not the only ones reported here. In control groups with no signs of autoimmune or inflammatory conditions, there is a complex network of connections between all subsets of regulatory B cells which, notably, are positive correlations only. In contrast, Graves’ disease in children was associated with pronounced changes in mutual relations among cells. It was demonstrated especially in the decreased population of CD38+Foxp3+IL-10+ Bregs that shifted dramatically into negative correlations versus other Breg subsets. It is the first time that combination of immunoregulatory CD38 and immunosuppressive IL-10 and Foxp3 markers in Bregs detection allowed the exposure of such essential disturbances in the cell connections network between healthy and Graves’ subjects.

Response to methimazole treatment was demonstrated to cause differential effects on immune cells closely related to the autoreactive reactions. It was reported that MMI application in Graves’ patients leads to an increase in initially reduced numbers of regulatory T cells, and reduction in Th17 cells elevated prior to the therapy [31]. Interestingly, as described in our study, such changes might predominantly be associated with the first period of methimazole use. In some subsets of Bregs, it led to a decrease towards pretreatment values in long-term observation. Such a trend for Bregs was observed in CD38+Foxp3+IL-10+ B cells with increased numbers only at the first 1–3 months of therapy. Noteworthy, subsets demonstrating the presence of Foxp3 and IL-10 but with no expression of CD38 responded to MMI therapy with elevated numbers, increasing the difference in reference to the healthy control group. These data support the idea of focusing on the outcome of the Bregs activity and not their phenotypic features only. More extensive population-based studies might reveal which of the changes observed are associated with restoration of proper immune balance and, thus, allow selection of Breg populations mainly engaged in the Graves-related phenomenon. Relatively worse remission rates are observed among children (25–29%) when compared to adults (40–75%) treated with methimazole; furthermore, 28% of pediatric patients demonstrated relapse after provisional remission [21,22]. Therefore, the question arises whether, here, reported changes in subpopulations of Bregs have a substantial role in response to methimazole, apart from already confirmed patterns of thyroid-related parameters (TSH, fT3, fT4, TRAb) linked to treatment efficacy [32]. As described above, methimazole use led to significant changes within Bregs population, both in their levels and their mutual correlations with thyroid-related parameters. However, we cannot reject the additional hypothesis that initial levels and function of regulatory B cells might affect efficacy of MMI application in the course of Graves’ disease. Further studies might answer that question and reveal the presence of an eventual relationship between Bregs and methimazole activity in the therapy.

Regarding correlation of studied Breg-related parameters with thyroid function, we found some essential links between selected subsets of B cells and TSH, fT3, fT4, and TRAb (TSH-R autoantibodies). We found, inter alia, that CD38+ B cell level is associated positively with levels of TRAb. On the other hand, they negatively correlated with TSH level prior to treatment application. To date, numerous studies associated CD38 presence with the development of autoimmune reactions and pathogenesis of such conditions as Sjögren’s syndrome, especially in the context of autoantibodies production [33,34]. Despite the fact of the correlation of CD38 with TSH-R antibodies observed here, we would more likely acquiesce to the hypothesis that CD38 is rather associated with regulation of Bregs function and immune response control, as suggested recently in a mice autoimmunity model [35]. That opinion is based on the reversed correlation of the CD38-negative B cell population when compared to CD38+ cells. Furthermore, the same dependencies were observed in TSH levels, where alterations in CD38 within the B cell population seem to be associated with initially low levels of that regulator of thyroid function. In addition, regulatory B cells with CD38 and Foxp3 expression also correlated with TSH, whereas CD38-Foxp3+ B cells were also closely related to values of fT3. Contrary to studies on other autoimmune conditions, such as rheumatoid arthritis, Foxp3+ and IL-10+ Bregs were not negatively associated with autoantibodies in Graves’ patients [36]. Furthermore, recently, negative links have been shown between TRAb and IL-10-producing Bregs [25] or CD1d+CD5+ B cells and thyroid-stimulating antibodies (TSAb) [37]. Here, we additionally found a correlation in the context of autoantibodies and B cells with IL-10 expression, although obtained values were characteristic for weak connections. These data do not exclude the essential role of IL-10 in Breg function; however, their contribution to the autoimmune process through suggested influence on autoantibody presence should be carefully considered. On the basis of the well-known essential role of Bregs in proper control of immune reactions, we assume that there is a significant link between these two populations. Furthermore, as B cells are the source of autoantibodies in the course of Graves’ disease, those mutual interactions might also affect thyroid gland function. Indicated aspects, however, require subsequent experiments aimed at the evaluation of exact interaction pathways leading to the phenomenon observed here in our study.

Importantly, observed correlations seemed to be strictly treatment-dependent, as methimazole application led to significant changes in connections described above. The role of CD38 presence was completely reversed, with a positive correlation of CD38+ with TSH and a negative for those B cell subsets lacking CD38. Interestingly, no relation to autoantibodies (TRAb) was reported after treatment. In addition, however, we found a strong/moderate association between CD38-positive Foxp3+ or IL-10+ Bregs and TSH levels. Comparable results were obtained in the context of CD38-negative Breg populations, and also CD38+Foxp3+IL-10+ cells, when analyzing correlations with values of fT4 in Graves’ patients treated with methimazole.

We are aware of the limitations associated with our study, predominantly related to the number of patients enrolled in the investigation. However, we applied strict selection criteria to work only with samples from the most representative subjects. Based on the predominantly very low p-values obtained, we presume that further increase in the sample size would only improve power of the test, but would not change the demonstrated findings. Nevertheless, further multicenter studies would be of great importance to enroll larger groups of patients and justify the results presented here on different subgroups of children. In addition, future projects on Bregs in Graves’ disease in pediatric patients could include more time points, and even extended periods of monitoring to verify more precisely any fluctuations within the studied population of cells in the course of MMI therapy.

Summary of changes in Bregs in the course of Graves’ disease—their association with thyroid-related parameters, and response to methimazole treatment application, provided a basis for further investigation of regulatory B cells’ initial levels influence on response to therapy. Despite the lack of essential differences at the beginning of treatment, we found that patients with higher numbers of CD38+Foxp3+ and CD38+Foxp3+IL-10+ Bregs demonstrated higher TSH levels achieved with methimazole use compared to subjects with low numbers of these subpopulations. Comparable results were shown in the context of autoantibodies for TSH-R, where their reduction was significantly effective predominantly in CD38+ B cells with Foxp3 or Foxp3 and IL-10 expression. Higher circulating numbers of B cells and increased tissue activity of these cells within the thyroid are a frequently reported phenomenon in Graves’ subjects [38]. In accordance, the acquired data demonstrating better reduction in TRAb and an increase in TSH can be justified with high CD38+ Breg levels and the related better control of immune response—peripherally and locally within the thyroid. Regarding Bregs influence on immune regulation and autoantibody level, high levels of CD38+Foxp3 and CD38+Foxp3+IL-10+ cells are likely to correlate negatively with TSH-R antibodies. A similar pattern was demonstrated in rheumatoid arthritis between IL-10-producing B cells and rheumatoid factor (RF) or anti-citrullinated protein antibodies (ACPA) [36]. Despite lack of correlations in untreated patients, we could presume the presence of a link between the tested Bregs and reported decline in TRAb, as demonstrated by other research groups recently [25,37]. Other crucial parameters, namely fT3 or fT4, did not show any significance between the Graves’ patient group stratified on the basis of selected Bregs numbers. Pretreatment values for fT4 in subjects with high levels of Bregs expressing Foxp3 seemed to be slightly higher. However, in the course of methimazole application, the final outcome for both groups was comparable. Similar data were obtained for patients with low levels of IL-10-expressing Bregs. In relation to IL-10+ Bregs, more attention should be paid to these subsets considering previous data on their essential function in immune response control and higher risk of autoimmune reactions in case of any disturbances in their function [39]. Here, we found that a more significant role could be attributed to regulatory B cells characterized by Foxp3 and IL-10 expression, as these subpopulations seemed to play the most crucial role in the course of Graves’ disease therapy. In general, our results indicate that Graves’ patients with higher initial levels of Bregs are predisposed for better response to the treatment, with proper values for TSH and efficient decline in TRAb within the first year of therapy. Further studies with long-term observation would be essential to validate the presented data and confirm the predictive value of the selected Bregs subsets in Graves’ pediatric patients’ management.

## 4. Materials and Methods

### 4.1. Patients

A total of 22 pediatric patients with active Graves’ disease (GD) were involved in the investigation. The control group (HC) consisted of 31 euthyroid healthy patients, demonstrating no signs of autoimmune and inflammatory conditions and no individual or family-related history of autoimmune thyroid disease (AITD). Peripheral blood was collected from Graves’s patients at three time-points: at the admission (before treatment/Time 0), and after 3 months (Time 1), and 1 year (Time 2) of the treatment implementation—methimazole. Patients with GD were treated with methimazole at an initial dose of 0.3–0.6 mg/kg/day in combination with propranolol at 0.5–1.0 mg/kg/day. Once clinical euthyroidism had been obtained (around Time 0 of the study), methimazole doses were reduced by 5–10 mg to reach the maintenance dose of 5 mg. Levothyroxine was added if hypothyroidism appeared.

An in-depth clinical description of the study subjects is included within the Appendix A. Written informed consent was obtained from each study subject (parent or legal guardian for underage patients). The Local Bioethical Committee approved the research protocol of the investigation at the Medical University of Bialystok (APK.002.78.2021).

Peripheral blood collected by venipuncture was subjected to gradient centrifugation using Pancoll with a density of 1.077g/l (PAN Biotech GmbH, Aidenbach, Germany) to obtain a fraction of peripheral blood mononuclear cells (PBMC). Subsequent washing of the cells in phosphate-buffered saline (PBS without Ca2+ and Mg2+; Corning) was followed by suspension in cryoprotectant (10% DMSO; Sigma-Aldrich, St. Louis, MO, USA) in fetal bovine serum (FBS; PAN Biotech GmbH, Aidenbach, Germany). PBMCs of the patients and controls were stored in liquid nitrogen to preserve viable cells until conduction of the described experiments.

### 4.2. Flow Cytometry

Thawing of PBMC stored in liquid nitrogen was followed by cell counting and viability verification. Subsequently, cells were stained with monoclonal antibodies conjugated to fluorochromes for flow cytometric analysis. Materials from Graves’ patients (GD) and control group (HC) were initially stained with a selection of antibodies aimed at cell surface markers, including anti-CD19 FITC (clone HIB19) and anti-CD38 PE-Cy5 (clone HIT2) (BD Bioscience, Franklin Lakes, NJ, USA). Investigation of absolute cell numbers was based on sample adjustment in the context of equal amounts of PBMC stained and the same buffer volume. Extracellular staining, after incubation and washing steps in phosphate-buffered saline (PBS; Corning), was followed by detection of intracellular markers—IL-10 and Foxp3, using: anti-IL-10 PE (JES3-9D7) and anti-Foxp3 AlexaFluo647 (259D/C7) (BD Bioscience, Franklin Lakes, NJ, USA). Following the incubation period and washing, cells were fixed using CellFix reagent (CellFix; BD Bioscience, Franklin Lakes, NJ, USA) and stored shortly in +4 °C until acquisition. FACS Calibur flow cytometer (BD Bioscience, Franklin Lakes, NJ, USA) was used to acquire data, analyzed subsequently using FlowJo software (Tree Star Inc., Ashland, OR, USA).

Delineation of regulatory B cells was established based on morphological properties (forward scatter (FSC)—relative size; side scatter (SSC)—relative shape/granularity), presence of CD19 for B cells, and expression of factors related to immunosuppressive function, namely IL-10 cytokine and forkhead box protein 3 transcription factor (Foxp3). In addition, evaluation of Bregs was supported by the use of CD38 marker—crucial in the context of cell proliferation, maturation, and activation. In accordance with the above information, several Bregs populations were distinguished: CD38-/CD38+Foxp3+, CD38-/CD38+IL-10, and those demonstrating simultaneous expression of IL-10 and Foxp3. The gating strategy implemented in the study was described in the Appendix A.

### 4.3. Statistical Analysis

Flow cytometric data were biostatistically processed using GraphPad Prism 9.0.0 statistical software (GraphPad Prism Inc., San Diego, CA, USA). Considering the non-normal distribution of the data within studied groups, two-way ANOVA with Fisher’s LSD test was implemented. The level of statistical significance was set at the value of < 0.05, with power highlighted with asterisks on the graphs: * *p* < 0.05, ** *p* < 0.01, *** *p* < 0.001, **** *p* < 0.0001. In addition, eventual tendencies for the difference between selected groups were described with the exact value of the *p*-value. The statistical analysis results are presented as mean values with standard deviations of absolute cell numbers and frequencies within CD19+ B cells. The assessment of correlations within studied regulatory B-cell-related parameters and clinical results was performed using the nonparametric Spearman test. Data are presented on the heat maps as correlation co-efficiencies (*r* values), and an asterisk highlights the presence of statistical significance. Depending on the r value, obtained results were considered as: weak (0.2–0.39), moderate (0.4–0.59), strong (0.6–0.79), or very strong (0.8–1.0) correlations.

## 5. Conclusions

Regulatory B cells play an essential role in the course of Graves’ disease. Despite numerous studies and a wide range of Breg subtypes investigated, our knowledge of their exact contribution and most significant phenotype is not complete. Here, we demonstrated for the first time that markers associated with immunosuppression do not seem to be strictly related to previously established phenotypes. Accordingly, we reported higher levels of Foxp3+ and IL-10+ Breg with CD38- phenotype, together with reduced numbers of the population most likely described to date as regulatory—CD38+Foxp3+IL-10+. With the use of CD38, Foxp3, and IL-10 to detect regulatory cells within the B cell population, we demonstrated essential differences in the correlation network within selected subtypes between Graves’ patients and healthy control subjects. Analysis of methimazole influence revealed most significant alterations within Bregs at the first 2-3 months of therapy. Only temporarily elevated subsets of CD38+Foxp3+IL-10+ B cells and gradually increasing Bregs with no CD38 expression allowed support of the previously drawn hypothesis. It stated that confirmation of B cells with regulatory function should be affirmed, focused more on the outcome of induced effects and not phenotypic features only. Regarding the study of immunosuppressive B cells’ association with thyroid function-related parameters, we found inter alia significant correlations of CD38-expressing B cells with TSH and TRAb levels. Importantly, treatment with methimazole influenced established links, with predominantly positive correlations of Breg parameters with TSH and fT4 reported. Finally, we demonstrated a better response to MMI therapy of Graves’ patients with higher initial levels of CD38+ Bregs expressing Foxp3+, with an efficient increase in TSH values and a decline in TSH-R autoantibodies. In conclusion, our data indicate a crucial role of regulatory B cells monitoring and their possible use as prognostic factors in Graves’ treatment management. Future investigations might validate the clinical utility of the Breg parameters described in our study.

## Figures and Tables

**Figure 1 ijms-22-10926-f001:**
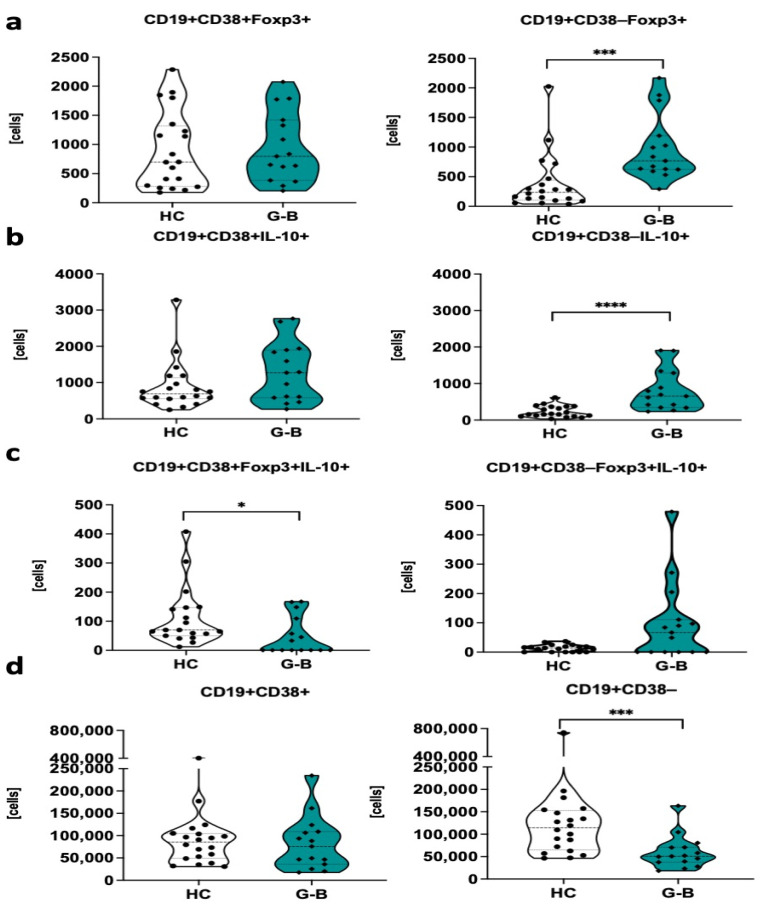
Differences among Graves’ disease patients and healthy control group in the context of regulatory B cell absolute numbers. Regulatory B cell subsets were distinguished based on differential CD38 marker expression and the presence of Foxp3 (**a**), IL-10 (**b**), co-expression of Foxp3, and IL-10 (**c**). Additional changes in CD38 expression within CD19+ B cells were included (**d**). Data are presented on each graph as mean with standard deviation. Significant data are indicated with asterisks: * *p* < 0.05, *** *p* < 0.001, **** *p* < 0.0001.

**Figure 2 ijms-22-10926-f002:**
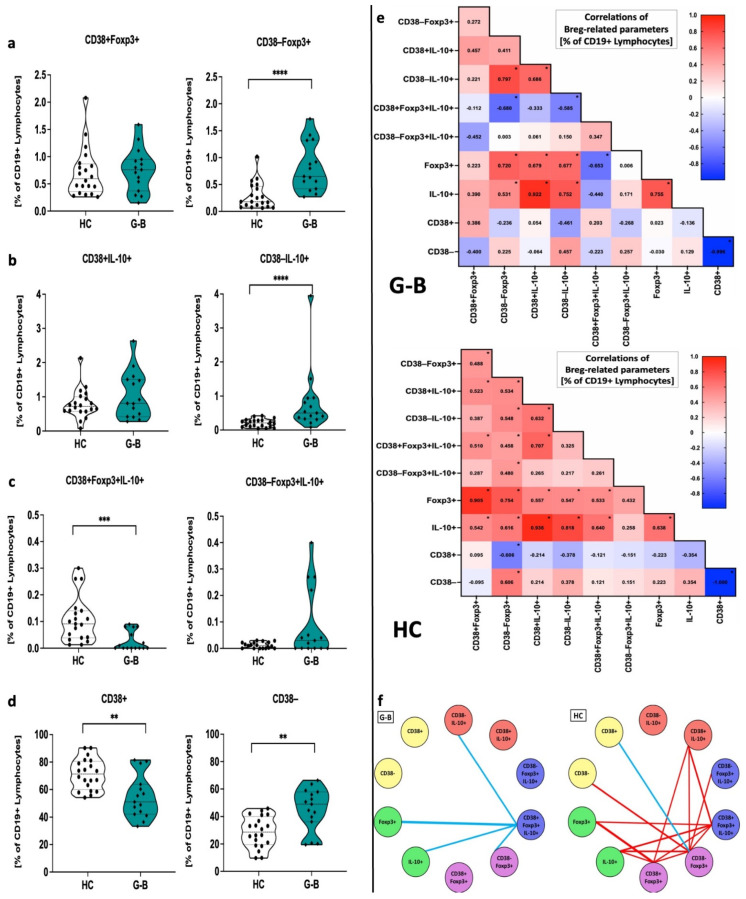
Variations in regulatory B cell frequencies in Graves’ patients and healthy control subjects. Regulatory B cell subsets were presented as B cells with/without the presence of CD38 marker and expression of Foxp3 (**a**), IL-10 (**b**), co-expression of Foxp3, and IL-10 (**c**). Changes in cells demonstrating CD38 within CD19+ B cells were also included (**d**). Data are presented on each graph as mean with standard deviation. Significant data are indicated with asterisks: * *p* < 0.05, ** *p* < 0.01, *** *p* < 0.001, **** *p* < 0.0001. Mutual associations between Breg-related parameters were established individually within Graves’ and healthy control subjects and presented as r values on heat maps (**e**). Significantly different connections between parameters within Graves’ and control patients were visualized following exclusion of overlapping links (**f**).

**Figure 3 ijms-22-10926-f003:**
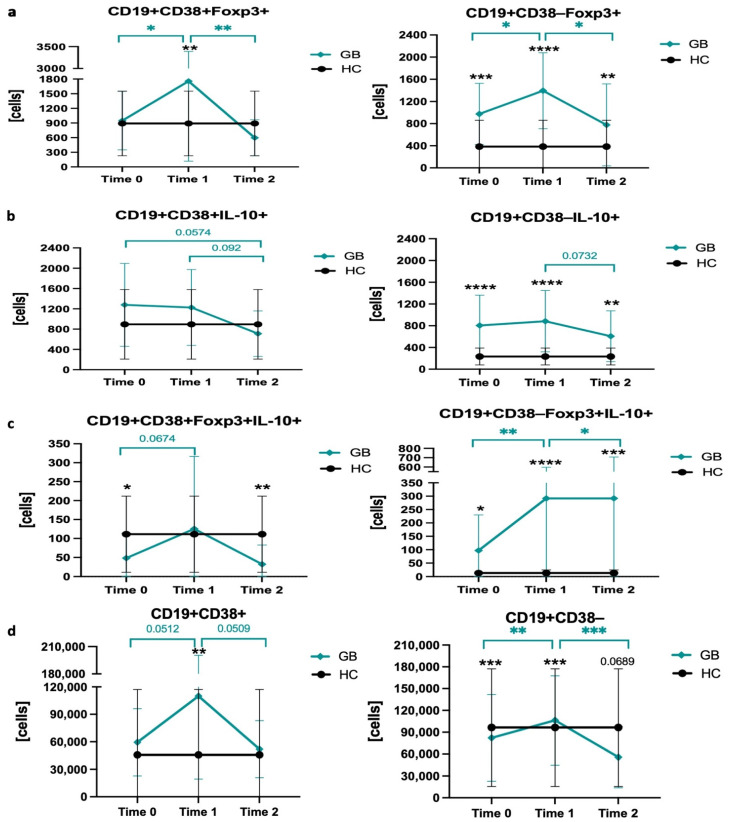
Methimazole treatment effects on distribution of regulatory B cells in Graves’ patients and healthy control group. Selected Breg populations were monitored for up to 3 months (Time 0) and then controlled after a 1–2-year interval (Time 2), with pretreatment values included (Time 0). Regulatory B cell subsets were distinguished based on differential CD38 marker expression and the presence of Foxp3 (**a**), IL-10 (**b**), co-expression of Foxp3 and IL-10 (**c**). Additional changes in CD38 expression within CD19+ B cells were included (**d**). Data are presented on each graph as mean with standard deviation. Significant differences are highlighted with asterisks or direct values: black—when comparing Graves’ to control group at a certain time point, and aquamarine—for changes in the course of therapy within Graves’ subjects. Significant data are indicated with asterisks: * *p* < 0.05, ** *p* < 0.01, *** *p* < 0.001, **** *p* < 0.0001.

**Figure 4 ijms-22-10926-f004:**
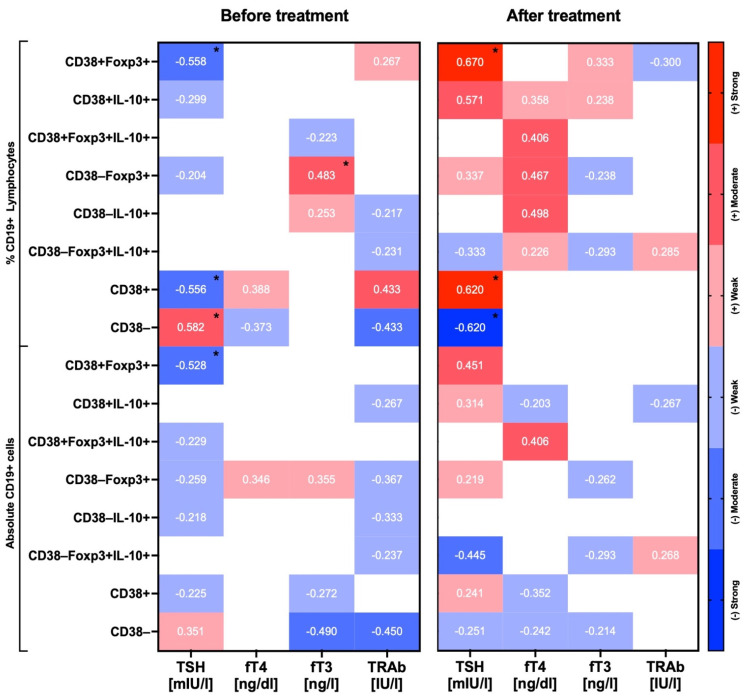
Correlations between essential thyroid function-related parameters and regulatory B cells in Graves’ patients. Links between studied parameters were established prior to and after treatment with methimazole. Heat maps present correlation co-efficiency values with those significant (*p*-value < 0.05), indicated by asterisks. Strength of correlations was demonstrated through color gradient—blue for negative correlations and red for positive correlations (*r* = −/+ 0.799–0.6—strong, *r* = −/+ 0.599–0.4—moderate, *r* = −/+ 0.399–0.2—weak correlations).

**Figure 5 ijms-22-10926-f005:**
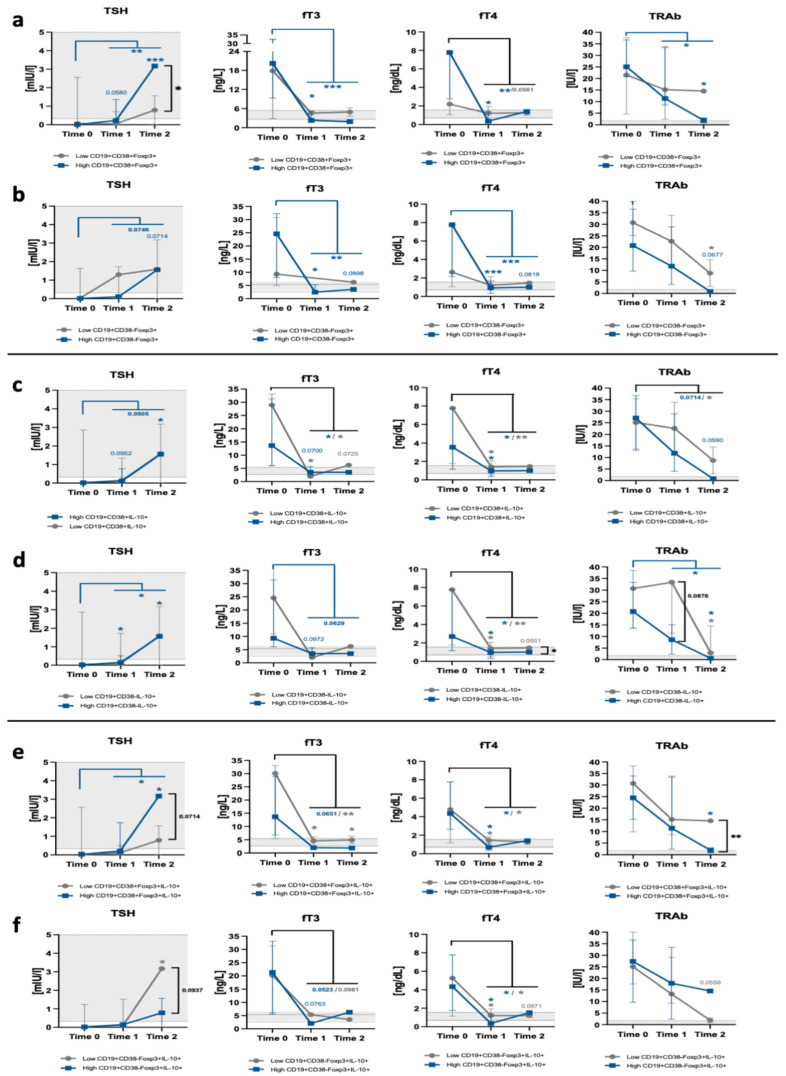
Monitoring of initial regulatory B cell levels’ significance in the context of Graves’ patients’ response to methimazole treatment. Graves’ subjects were stratified into low- (grey lines) and high-Breg (blue lines) groups and monitored at selected time points for thyroid-related parameters, namely TSH, fT3, fT4, and TRAb. Low/high stratification was applied to selected Breg populations: CD19+CD38+Foxp3+ (a), CD19+CD38-Foxp3+ (b), CD19+CD38+IL-10+ (c), CD19+CD38-IL-10+ (d), CD19+CD38+Foxp3+IL-10+ (e), CD19+CD38-Foxp3+IL-10+ (f). Statistically significant differences in parameters versus these observed prior treatment applications are indicated with an asterisk, with a color corresponding to the specific group. When needed and to support conclusions made, a general before versus after treatment comparison was also performed. * *p* < 0.05, ** *p* < 0.01, *** *p* < 0.001.

## Data Availability

The data presented in this study are available on request from the corresponding author.

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
