# Peer review of "Regulatory B Cells Involvement in Autoimmune Phenomena Occurring in Pediatric Graves’ Disease Patients"

_ijms, 2021, doi:10.3390/ijms222010926_

Round 1

Reviewer 1 Report

This is a very interesting and well prepared paper, presenting a significant role of regulatory B cells in Graves' disease and their possible application as prognostic factor in the management of the disease.

I have only a few minor remarks:

  1. The analyzed group of children with Graves' disease is quite small and the obtained results require further confirmation in a larger group of patients. Please, include a paragraph with study limitations and discuss them shortly.
  2. Line 99 – please indicate that the Time 1 is after 3 MONTHS
  3. Please provide an English language correction made by a native speaker

Reviewer 2 Report

Grubczak et al. present an investigation of regulatory B cell populations in pediatric Graves' patients with three aims: 1) compare and contrast B cell populations in pediatric Graves' patients and healthy controls; 2) detect changes in Bregs following MMI treatment; and 3) assess the possible relationship between Breg population and biomarkers of thyroid function. 

From a scientific perspective, this paper's greatest weakness lies in the attempt to assign biological significance to statistical significance. I specifically refer to the correlations between Breg subtypes and thyroid functional markers. Are the authors proposing a causative relationship? If so, which direction? What further confuses this point is that the authors propose, in their introduction, that MMI treatment acts on Breg populations. Yet when describing their results, lines 273-275, the authors suggest that the pre-existing Breg subpopulations affect the efficacy of MMI treatment. This jump in logic requires further explanation.

Also from a scientific perspective, the authors need to further justify the use of a pediatric population in this investigation. Despite the presence of "pediatric" in the title, the unique characteristics of these patients compared to adult Graves' patients are not described. Why is it necessary to separately study pediatric patients rather than rely on studies with adults? This question should be addressed either in the Introduction or Discussion.

From a reader's perspective, the science is almost completely obscured by the complex sentence structure and word choice. For example, lines 307-309,

Multiple subtypes distinguished and differential functions of Bregs reported cause a situation in which they still constitute unresolved question in context of their exact role in autoimmunity.

This sentence has multiple possible subjects and verbs. Different readers would interpret this sentence in different ways. Converting this sentence into plain language would reduce the possible meanings down to the one the authors intend to communicate.

This problem persists throughout the paper and prevents the reader from appreciating the data. The readability of the prose must be improved before this manuscript is ready for publication.

Round 2

Reviewer 2 Report

My concern about the readability of the text has not been addressed. Below I provide specific examples in the Results section that need to be changed in order to understand how the authors intended to present the data. Please keep in mind, some of the figures are complex. Balancing that complexity with concise descriptions in the text would help people appreciate the data.

First sentence of Results, change "studied groups" to "healthy vs Graves."
"studied groups" is problematic, because you are in truth making two comparisons, healthy vs Graves and C38+ vs CD38-. Explicitly stating what "studied groups" are, at least once, leaves no room for confusion.  

Third sentence, first paragraph of Results, change "Interestingly, however" to "Compared to Healthy Controls, pediatric Graves' patients showed..." and modify the rest of the sentence as necessary.
The original wording of this paragraph implies that the authors are describing Graves' results compared to healthy controls. Explicitly stating this comparison cuts down on possible confusion.

"In addition, described changes were concomitantly followed by a reduced number of CD38- B cells"
Which described changes are the authors referring to? A majority of readers would attribute "described changes" with the decline in CD19+CD38+IL10+Foxp3+ because that was the most recent change mentioned. How is this naturally accompanying or associated with a reduction in CD38- cells? Frankly, I would avoid use of "concomitantly" in the Results section because it is an interpretation of the data. This whole problem would be avoided by simply stating something like, "In addition to described changes, pediatric Graves' patients showed a decrease in C19+CD38- cells."

Also in the first paragraph of Results, include references to Fig 1a, 1b, and 1c when mentioned in the text. That would also cut down on possible confusion.

Second paragraph of Results, first sentence, "obtained data are in strict accordance with absolute values"
Please clarify in the text what makes this accordance strict. Are these values exactly the same? Are they in direct proportion to? Alternatively take out "strict" and use "in accordance" or "in agreement," which would avoid the problem in the first place.

Second paragraph of Results, third sentence, "Moreover, reduced values for CD38+ cells with co-expression of...in Graves' disease pediatric patients" needs a verb for the main clause.

I do appreciate how the text in this paragraph explicitly states the healthy vs Graves' comparison.

Third paragraph of Results, first sentence, does "reported differences" refer to differences described in previous publications or the differences described immediately above? If the former, include a citation. If the latter, change "reported" to "above." Alternatively, remove the entire subordinate clause and start the main clause with something like "We further wondered..." 

Third paragraph of Results, fifth sentence, "Moreover, these subsets showed..."

Remove, "...thus, completing described differences between Graves's [sic] patients and healthy controls." It is confusing to say your description is complete and then go on to describe additional differences for several more sentences.    

Fourth paragraph of Results, include references to Fig 3a, 3b, 3c, and 3d when they are mentioned in the text.

Fourth paragraph of Results, second sentence, "In most cases, crucial had appeared already at a first time point, namely up to 3 months of therapy." I do not know what this means, please clarify.

Fourth paragraph of Results, fifth sentence, "In the context of IL-10-producing CD19+C38-/+ cells, their number remained comparable to this prior therapy, with only a slight tendency for values comparable to healthy subject after 1-2 years of MMI."
The wording is confusing. Cell number before and after MMI was unchanged? Cell number was comparable to a different prior therapy? Does comparable mean not statistically significant? Does "slight tendency" mean a non-statistically significant change over time? Is the subordinate clause supposed to say, "...for values to become comparable?"

Fifth paragraph of Results, fourth sentence, "In contrast, a negative association was found between CD38-negative cell numbers and fT3, additionally, with a moderate positive correlation of that parameter with the frequency of CD38-Foxp3+ in B cells pool." Change "negative association...CD38-negative cell numbers..." to "positive association...CD38+ cell numbers."
The original wording is tricky because the previous sentence describes a negative correlation with TSH vs CD38-, while this sentence describes a negative correlation with fT3 vs C38+. The reader is left to assume that if the negative correlation is statistically significant for CD38+ cells, then a positive correlation with CD38- cells would also be statistically significant, and vice versa. This is not always the case. The sentence would be easier to interpret if the subjects of both sentences were CD38+ cell number.

Or did I misinterpret the sentence and "In contrast" refers to the fact that TSH was highly negatively correlated with both C38+ and CD38- (this interpretation based on the fact that p values for both CD38+ and C38- cells were mentioned in the text), whereas fT3 negatively correlated with one thing but positively correlated with another?

Sixth paragraph of Results, fifth sentence, "Interestingly, the observed...the intensity of changes in TSH was reversed..."
How was the intensity reversed? Were the changes initially high in magnitude, them became low in magnitude? Did the direction of the trend from one time point to another go in the opposite ways?

First paragraph of Discussion, third sentence "...cause a situation in which they still constitute unresolved question [sic] in the context of their exact role in autoimmunity." has too many subordinate clauses. An example of a concise sentence would be, "Our effort to understand the Bregs' role in autoimmunity is hindered by it's multiple phenotypes and functions." 

Round 3

Reviewer 2 Report

I appreciate your effort.